# Expanding Understanding of Urban Rift Valley Fever Risk and Associated Vector Ecology at Slaughterhouses in Kisumu, Kenya

**DOI:** 10.3390/pathogens13060488

**Published:** 2024-06-08

**Authors:** Keli Nicole Gerken, Kevin Omondi Owuor, Bryson Ndenga, Sammy Wambua, Christabel Winter, Salome Chemutai, Rodney Omukuti, Daniel Arabu, Irene Miring’u, William C. Wilson, Francis Mutuku, Jesse J. Waggoner, Benjamin Pinsky, Carren Bosire, Angelle Desiree LaBeaud

**Affiliations:** 1Department of Pediatrics, Division of Infectious Diseases, Stanford University School of Medicine, Stanford, CA 94305, USA; bpinsky@stanford.edu (B.P.); dlabeaud@stanford.edu (A.D.L.); 2Kenya Medical Research Institute, Centre for Global Health Research, Kisumu 40100, Kenya; owuorkevinomondi@gmail.com (K.O.O.); bndenga@yahoo.com (B.N.); christabelwinter@gmail.com (C.W.); 3Pwani University Biosciences Research Centre (PUBReC), Pwani University, Kilifi 80108, Kenya; sammywambua@gmail.com (S.W.); salomechemutaic@gmail.com (S.C.); omukutirodney@gmail.com (R.O.); dantearabu@gmail.com (D.A.); miring.irene@gmail.com (I.M.); 4Research and Conservation Support Society (RECOURSE), Kilifi 80108, Kenya; 5School of Biodiversity One Health and Veterinary Medicine, University of Glasgow, Glasgow G12 8QQ, UK; 6Foreign Arthropod-Borne Animal Disease Research, United States Department of Agriculture-Agriculture Research Service (USDA-ARS), Manhattan, KS 66502, USA; william.wilson2@usda.gov; 7Department of Environment and Health Sciences, Technical University of Mombasa, Mombasa 80110, Kenya; fmutuku73@gmail.com; 8Division of Infectious Diseases, Department of Medicine, Emory University School of Medicine, Atlanta, GA 30322, USA; jesse.waggoner@emoryhealthcare.org; 9Department of Pure and Applied Sciences, Technical University of Mombasa, Mombasa 80100, Kenya; crrnbosire@gmail.com

**Keywords:** Rift Valley fever, epidemiology, urban zoonoses, slaughterhouse surveillance, bloodmeal analysis

## Abstract

Rift Valley fever virus (RVFV) is an adaptable arbovirus that can be transmitted by a wide variety of arthropods. Widespread urban transmission of RVFV has not yet occurred, but peri-urban outbreaks of RVFV have recently been documented in East Africa. We previously reported low-level exposure in urban communities and highlighted the risk of introduction via live animal influx. We deployed a slaughtered animal testing framework in response to an early warning system at two urban slaughterhouses and tested animals entering the meat value chain for anti-RVFV IgG and IgM antibodies. We simultaneously trapped mosquitoes for RVFV and bloodmeal testing. Out of 923 animals tested, an 8.5% IgG seroprevalence was identified but no evidence of recent livestock exposure was detected. Mosquito species abundance varied greatly by slaughterhouse site, which explained 52% of the variance in blood meals. We captured many *Culex* spp., a known RVFV amplifying vector, at one of the sites (*p* < 0.001), and this species had the most diverse blood meals. No mosquito pools tested positive for RVFV antigen using a rapid VecTOR test. These results expand understanding of potential RVF urban disease ecology, and highlight that slaughterhouses are key locations for future surveillance, modelling, and monitoring efforts.

## 1. Background

Rift Valley fever virus (RVFV) is an extremely adaptable arbovirus that has recently been demonstrated to have more potential host–vector interactions than many other viruses of medical importance [1]. It is endemic across all regions of the African continent and sporadically causes explosive outbreaks linked to flooding. During these outbreaks and interepidemic transmission events, domestic ruminants, including cattle, sheep, goats, and camels, are the most impacted species [2]. This impact extends into public health risk when humans carry out activities that expose them to infected livestock fluids [3]. Slaughterhouse workers and those with significant animal contact are often the first observed human cases after spillover, as demonstrated in the most recent outbreaks in Uganda [4]. Beyond this direct occupational exposure risk, the relative risk of infection from mosquito bites has not been quantified in this setting or in the general community of endemic countries. A peri-urban outbreak of RVFV in livestock was retrospectively confirmed in Northern Tanzania recently, as one of the first confirmed outbreaks of RVFV with urban involvement [5]. Our research team has been examining this question in the context of urban Kenya through a series of One Health studies. Our findings to date include documentation of low-level risk in the general community independent of livestock ownership [6], a qualitative investigation of factors that could influence the introduction and persistence of RVFV in urban settings [7], and the feasibility of slaughterhouse-based livestock surveillance highlighting movement from markets to be key for determining animal origin [8]. However, the role of arthropod vectors at urban slaughterhouses or in other places where animals congregate remains undefined. Thus, a full understanding of the life cycle and potential urban disease ecology of RVFV remains unknown.

Vector trait dynamics drive transmission potential for most vector-borne diseases [9], yet, assessing transmission pathways of RVFV to humans is confounded by the direct transmission pathway. However, in livestock, risk of infection is directly linked to their environments and the arthropods within these, as livestock are highly inefficient at horizontal transmission even when their immune systems are iatrogenically suppressed [10]. RVFV can be transmitted by a plethora of different arthropods, yet, mosquitos, by and large, represent the most common vectors [2,11]. Despite this importance, assessing overall vectorial capacity for RVFV is complicated by the varying competence of many different mosquito species and that even during the same outbreak, different species can dominate transmission between geographical areas [12]. *Aedes* floodwater mosquitos have been classically described as the ‘primary RVFV vectors’ that are capable of laying infected eggs that remain dormant and viable until emergence during floods to initiate outbreaks [13]. Thereafter, it has been proposed that ‘secondary vectors’ such as *Culex* and *Mansonia* spp. amplify RVFV transmission [12]. Vertical transmission has also recently been demonstrated in the laboratory with *Culex tarsalis* [14]. However, capturing the timing of these vector roles in transmission has been challenging in field conditions and roles may shift depending on ecological conditions and the level of RVFV endemicity [15].

The urban-to-rural ecological spectrum is a key interface that influences vector populations. Many studies conducted on urban vector-borne diseases have focused on urban malaria transmission, demonstrating heterogenous inoculation rates between and within cities leading to differential risk of vector-borne disease [16]. Other arboviruses, including dengue and yellow fever viruses, have established urban adaptation mechanisms that have allowed them to expand and cause larger, more devastating outbreaks [17]. RVFV vectors have not been assessed directly in the urban context and extrapolating risk based on other vector studies does not capture the diverse environments to which livestock, and subsequently humans, can be exposed in the urban environment beyond the household level. Conducting an analysis of all competent vectors and all vectors present in urban/peri-urban environments could, in theory, build a basis to understanding all potential host–vector interactions, but this is likely to have little relevance to transmission dynamics at a specific urban site.

*Culex quinquefasciatus* has been previously described as an important amplifying vector and successful transmitter of RVFV in East Africa [18]. Another study that used diverse ecological modeling described very suitable distributions for the RVFV vectors *Aedes aegypti* and *Culex pipiens* in East Africa, with Northwestern Tanzania as the most profound hotspot for these species [19]. Studies such as that provide a guide to understanding the many RVFV vector–host interactions, but the complexity of RVFV transmission requires that our understanding of RVFV also be highly dynamic. Assuming that data can be extrapolated from one ecological condition to another may not be wise, particularly in diverse urban environments.

RVFV outbreak potential is modulated by several factors independent of vector abundance, including livestock density, livestock movements, and baseline seroprevalence, which is influenced by RVFV activity in prior years [11]. A study in Northeastern and coastal Kenya mapped mosquito species richness and found significant differences in RVFV abundance and diversity of RVFV vectors along movement routes. These differences were thought to be driven by the ecological zone of the trap site [20]. Indeed, the overlap with livestock movement risk and ecological conditions complicates disentangling if RVFV outbreaks are initiated through emergence of mosquitos that were dormantly infected eggs or infected livestock movement.

Nearly all systematic reviews of RVFV epidemiology have pointed towards inconsistent data collection, both temporally and spatially, being a key reason that gaps in RVFV epidemiology persist, such as the dominant vectors and viral maintenance mechanisms [11,21,22]. In some locations, new potential host species may also become important, as previously suggested by the theoretical transmission of RVFV to North American white-tailed deer among other hoofed ungulates [23,24]. The norms for livestock movement in these hypothetical settings are vastly different to movements in RVFV endemic countries such as Kenya. Another recent study documented a high risk of human exposure among movement routes and this provides further evidence that livestock walking routes are an important mechanism for seeding, or introducing, RVFV to naïve areas [25]. Transmission could additionally be sustained by livestock movements even in the absence of new introduction events, as demonstrated in a model of livestock being transported between the Comoros Islands [26]. As RVFV becomes more endemic and occurs more frequently, the ability to recognize clinical signs in affected animals is likely to wane further as populations have different sizes and higher baseline prevalence [27]. It will be more challenging to detect sick adult animals without systematic surveillance, and this continues to be undeniably linked to our lack of understanding of basic transmission principles for RVFV.

The correlation of mosquito abundance and weather patterns has led to many efforts to leverage these associations and predict heightened risk of RVFV outbreaks to guide implementation of surveillance and preventive measures. A Food and Agriculture Organization (FAO) early warning system triggered an alert for RVF in Eastern Africa on 16 February 2022. Here, we present data collected during this high-risk time when our previously described testing framework at slaughter was expanded to include another urban slaughterhouse and acute RVFV diagnostics. Additionally, we build an understanding of amplification risk in livestock at urban slaughterhouses through describing urban vector ecology and host bloodmeal preferences. The methods described here are designed with the intention that they can be used in different urban conditions where livestock congregate to characterize risk and parametrize models to assess amplification potential.

## 2. Methods

### 2.1. Study Overview Summary

In this study, we initiated sampling of livestock blood at two slaughterhouses, Mamboleo and Rabuor, in April 2022, six weeks after the FAO regional early warning alert. The following month, we added a vector component and trapped mosquitos to describe vector ecology and bloodmeal seeking patterns. Livestock serum samples from slaughterhouses were tested for RVFV prior exposure (IgG antibodies) and recent exposure to RVFV (IgM antibodies). We also tested blood fed and gravid mosquito pools using a lateral flow assay for RVFV antigen (VecTOR Test Systems, Inc., Thousand Oaks, CA, USA, www.vectortest.com last accessed on 7 June 2024) and carried out bloodmeal analysis on a subset of representative mosquito pools using the metabarcoding methods described below. All field operations commenced on 1 July 2022.

### 2.2. Surveillance of Livestock Blood

We utilized the same sampling framework described in Gerken et al. 2022 [8]. In brief, this integrative surveillance system relies on individuals already working at the slaughterhouse to carry out sampling though a process in which the slaughterman is passed an empty 15 mL conical tube to fill with blood directly after slaughter and he identifies the owner of the specific animal. An assistant is then directed to the appropriate business stakeholder and orally delivers a brief survey on a laminated card including the animals’ origin, transport means, herd size, holding time for slaughter, and age estimated by dentition once the animal’s head is removed by the slaughterman. In this study, we aimed to adapt the sampling framework to another slaughterhouse with a different layout, equipment, and flow of animals.

### 2.3. Study Site Description

The study was carried out in Kisumu City at two main slaughterhouses serving the urban meat value chain. Briefly, Kisumu is the third largest city in Kenya and is located on the Eastern bank of Lake Victoria. Kisumu city has been described in our other studies [6,7] in greater depth and here, we focus on the differences between the two slaughterhouse sites.

The two participating slaughterhouses that serve the urban market in this study were Mamboleo and Rabuor. Mamboleo slaughterhouse is the largest slaughterhouse in Western Kenya and has a cattle chute/crush for slaughter, concrete floors, and a hydraulic lift to invert the animals before manual slaughter (cutting of the neck) by a Muslim slaughterman. Animals are then lowered to the concrete flooring of the slaughterhouse where the head is removed, a midline cut made, and skinning takes place on the slaughterhouse floor. Cattle and small stock are housed separately overnight at this slaughterhouse, which is constantly resupplied each day with new animals. Neither a crush nor a captive bolt is used for small stock (sheep and goats) and instead, an employee manually brings them one by one from the holding pen to the slaughter slab.

In contrast, the Rabuor slaughterhouse does not have holding facilities for animals awaiting slaughter and was previously reported to contract nearby households to care for the animals for up to two weeks while they await slaughter [7]. Rabuor slaughterhouse does not have a cattle chute/crush, captive bolt gun, or hydraulic lift. Instead, animals are slaughtered on arrival after transport by various stakeholders all throughout the morning hours. The animals are immobilized, and slaughtering occurs on a concrete slab outside the slaughterhouse door. The animal is then dragged into the slaughterhouse for skinning and filleting using a pully system lift. All carcasses and associated organs remain hanging until the meat inspector veterinarian has given their stamp of approval. Blood is also collected in plastic buckets from the exsanguination process to await informal trading at the end of the slaughtering day.

### 2.4. Laboratory Methods for Livestock Blood Screening

Individual serum samples were tested for anti-RVFV IgG and IgM antibodies using commercialized kits from ID Vet (Grabels, France). Serum was separated from blood clots on the same day they were collected, and we followed the manufacturer’s protocol verbatim. As per the protocol, plates were read at 450 nm and results were calculated and interpreted according to the cutoffs described in the protocol.

### 2.5. Mosquito Vector Trapping

We deployed ovitraps and Biogents (BG) traps and recruited an experienced Prokopack aspirator user. This same individual sorted and morphologically identified all vectors captured in this study. Eight ovitraps [28] were set on Mondays outside the area surrounding the main slaughterhouses and were collected on Fridays. The paper inside was retrieved, wrapped in filter paper, put inside a cooler box, and taken to the laboratory to count the *Aedes* eggs using a 10× dissecting microscope; these results were recorded on a data form. The ovitrap status at the time of collection was recorded either as good, water-reduced, dry, fallen, or lost.

One Biogents (BG) trap (Biogents AG, Weissenburgstr 22, 93055 Regensburg, Germany) baited with carbon dioxide (CO_2_) was set Monday–Friday on alternate weeks at each of the slaughterhouses [29]. At Rabuor, the BG trap was set inside the slaughterhouse with its half-top wall open but covered by a two-inch metal wire mesh as there was no security at night to guard the trap. At Mamboleo, the BG trap was set adjacent to the cattle holding shed during week one of sampling. For the following weeks, a wooden frame secured with a wire mesh was constructed to protect the trap, which was placed in the center of the cattle holding pen. The BG trap was held in this wooden frame until the final week of sampling, whereupon a bull destroyed the construction, and we elected not to rebuild it. In the final week of sampling, the BG trap was outside of the cattle enclosure. Trapped mosquitoes were collected daily between 9:00 a.m. and 12:00 p.m. from Tuesdays to Fridays and identified in an allocated field office space onsite at the slaughterhouses. The BG trap battery was replaced daily.

Sampling using automated Prokopack aspirators was conducted on alternating weeks at each slaughterhouse inside the main slaughterhouse building (indoors) and the outdoor surroundings within its compound for ten minutes each between the hours of 8:45 a.m. to 12:00 p.m. [29]. Two pre-labelled plastic cups were used to capture mosquitoes, one indoors and the other outdoors, to sample all corners and places with less human interference where mosquitoes rested during the day. Collected mosquitoes were killed using a pyrethrum aerosol spray before they were identified and recorded.

### 2.6. Acute RVFV Detection: Lateral Flow Assay and Confirmatory PCR

We separated female mosquitoes according to their blood-feeding stages as unfed, blood-fed, half-gravid, or gravid. These mosquitos were preserved in silica gel self-indicating 6–20 mesh (Blue)—500 gm (Loba Chemie Pvt. Ltd., Jehangir Villa, 107, Wodehouse Road, Colaba, Mumbai 400 005, India) for further testing in the laboratory using a lateral flow assay developed for RVFV antigen detection in mosquitos (VecTOR test). The development of this wicking lateral flow assay was modeled on the VecTOR test for West Nile virus. It has previously been described to have a high specificity (99.6%) to detect RVFV antigens in mosquitoes [30]. A positive result was visually interpreted through the presence of two lines vs. one control line.

### 2.7. Real-Time Polymerase Chain Reaction (RT-PCR) to Confirm Lateral Flow Assay Results

All suspect positive lateral flow tests were confirmed via real-time PCR (RT-PCR) using protocols developed in the Pinsky laboratory at Stanford University. RNA was extracted from either the mashed mosquito homogenate or serum using a Thermo Scientific™ GeneJET RNA Purification Kit (Thermo Scientific™, Waltham, MA, USA).

### 2.8. Description of RT-PCR Assay for RVFV

Our reverse transcription quantitative real-time polymerase chain reaction (RT-qPCR) assay was developed for the detection of RVFV in human samples (Pinsky laboratory, Stanford University). The purpose of this test is to provide sensitive RVFV RNA detection in clinical specimens and provide the potential for multiplexing with other targets. RVFV-specific primers and a hydrolysis probe were designed for the small genome segment (S), optimized for sensitivity, and used in the final reaction mixture at 200 nM each. Final primer and probe sequences were as follows 5′-3′: GATTTGCAGAGTGGTCGTCG (RVFV_S_F), CGATGGTGCATGAGAAAGACA(RVFV_S_R), FAM-ACCTTATTCTATGGTTGGGCCCTGT-BHQ1 (RVFV_S_Probe).

The assay can demonstrate linear detection from 2.0 to 8.0 log_10_ copies/µL of eluate and provide specific detection when tested against common arboviruses. This assay was developed for RNA extraction using a manual nucleic acid kit from Thermo Scientific (GeneJET RNA Purification Mini Kit), utilizing silica column-based purification technology for the isolation of highly purified nucleic acids. Following RNA extraction, 10 μL of sample nucleic acid and 15 μL (primer/probe/enzyme master mix) were loaded into a magnetic induction cycler (MIC qPCR; BioMolecular Systems) with the following cycling conditions: 52 °C for 15 min; 94 °C for 2 min; 45 cycles at 94 °C for 15 s, 55 °C for 20 s (acquisition), and 68 °C for 20 s. Each run included a no-template control and a positive RVFV control (synthesized single-stranded DNA (ssDNA) containing the target sequence).

### 2.9. Mosquito Bloodmeal Analysis in the Laboratory

Samples of blood-fed mosquitoes were transported on dry ice to Pwani University Bioscience Research Centre (PUBReC) in Kilifi, Kenya, for bloodmeal analysis. Each tube had 1–6 mosquitoes in total and out of these, 14 pools were made based on mosquito species and collection site. Genomic DNA extraction was carried out on individual mosquito abdomens using the TIANamp Genomic DNA Kit (Tiangen, Beijing, China) following the manufacturer’s protocol with some modifications. Modifications involved reducing lysis buffer to 150 µL, proteinase K to 15 µL, vortexing during lysis, and lysis time to 30 min. The quality and quantity of DNA was analyzed using a NanoDrop 2000C spectrophotometer (Thermo Scientific Inc., Waltham, MA, USA) and verified on 1% agarose gel electrophoresis. The DNA samples were then batched into 14 pools each of ~3.14 mosquitoes (SD = 1.46). To identify bloodmeal sources, ~300 base pairs (bp) of the cytochrome b barcode were PCR-amplified [31] and subjected to high-throughput-sequencing at Macrogen Inc., Seoul, Republic of Korea, using the Illumina 300 × 2 bp platform (Illumina, San Diego, CA, USA).

### 2.10. Statistical Analysis

Livestock serum samples were analyzed with the individuals’ IgG antibody status as a binary outcome (positive or negative). Data were initially examined using histograms, bar plots, and boxplots to observe the distribution of the data. Descriptive statistics were computed and where appropriate, *t*-tests, Chi-square tests, and simple logistic regression were performed to examine the statistical significance of relationships between predictor variables and the outcome (IgG antibody status). To compare differences in vector data between the two slaughterhouses, an analysis of variance (ANOVA) was computed. Analysis of livestock data was carried out using R version 4.1.2 [32] and mosquito data analysis was carried out using Statistical Package for the Social Sciences (SPSS) version 20 (SPSS, Chicago, IL, USA).

### 2.11. Analysis of Mosquito Bloodmeals

Sequences were analyzed via two approaches, for taxonomical annotation. The first approach employed an operational taxonomic unit (OTU)-based pipeline, USEARCH v11.0.667 [33]. Here, sequences were trimmed and filtered using TRIMMOMATIC v0.39 [34] and pre-processed with the standard USEARCH commands. Paired end reads were merged using *usearch –fastq_mergepairs*, followed by an additional step for quality filtering using VSEARCH v2.24.0 [35] with the command *–fastq_filter*. The merged reads were then dereplicated using *usearch –fastx_uniques* and OTUs clustered at 97% sequence identity using the command *usearch –cluster_otus*. Annotations were carried out with the *usearch –otutab* function and taxonomy assignments initially conducted with *usearch –usearch* against the MIDORI reference database, containing mitochondrial DNA sequences from eukaryotes, after downloading cytochrome b sequences vGB257 in the SINTAX format, and then, against the NCBI database using the command *blastn* [36] with a ≥99% identity threshold for species assignment. The second approach utilized DADA2 v1.13.0 [37], an amplicon sequence variant (ASV)-based package in the R programming environment. Raw reads were trimmed and filtered according to sequence quality and length. After dereplicating and implementing the error learning model, ASVs were inferred. Taxonomy was assigned via exact matching against DADA2-trained MIDORI2 reference database vGB254, which included eukaryotic mitochondrial sequences [38]. Where identities from the different assignments did not match, the lowest common taxonomic level was reported.

All statistical analyses for bloodmeal analysis were conducted in R v4.1.2 [32]. Rarefaction curves and alpha diversity metrics, e.g., species richness and relative abundance, were estimated with the *vegan* package v2.6.4 [39] and plotted using *ggplot2*.

### 2.12. Ethics

For the livestock sampling component of this study, we followed the same ethical approval protocol described in the pilot study at the second slaughterhouse [8]. Only animals from business stakeholders that signed a consent form were included in our study. For vector trapping, we informed slaughterhouse employees, management, livestock owners, and meat inspector veterinarians about the goals of our projects. The entirety of this study was approved by the ethical review board at Stanford University (IRB protocol # 61386) and Kenya Medical Research Institute (KEMRI/SERU/CGHR/03-07-390/4293). A research permit (License No: NACOSTI/P/21/13557) to conduct this study was obtained for the National Commission for Science, Technology and Innovation (NACOSTI) in Kenya.

## 3. Results

### 3.1. Livestock Serum Testing

In total, over the course of two months, we collected serum from 923 individual animals at slaughter and tested them for anti-RVFV IgG and IgM antibodies. We did not detect any positive IgM antibody samples and recorded an overall IgG seropositivity rate of 9% (78/923), indicating that while animals had not recently been infected, these urban slaughterhouses were indeed slaughtering animals that had recovered from RVFV infection previously. Seropositivity was significantly higher in cattle (20%; 64/325), compared with goats (2%; 6/285) or sheep (3%; 8/313, *p* < 0.001). There was no significant difference in seroprevalence between the two slaughterhouses (*p* = 0.90), Mamboleo (8%, 42/503) and Rabuor (9%, 36/420) that serve the urban meat market (Table 1).

Animals waited between 1–15 days to be slaughtered (mean = 2.4), and all seropositive animals were slaughtered within eight days of arrival. Nearly half (45%) of animals were slaughtered within one day of arrival, 19% within two days, and an additional 16% within three days. Herd sizes ranged from 1–100 and smaller herd sizes were associated with seropositivity (*p* = 0.02).

We sampled for a total of 13 weeks with 64 sampling days at the two slaughterhouses, starting with Mamboleo and alternating to Rabuor every other week. On average, we collected 14 samples per day (range = 10–18). The sampling day with the highest rate of seropositivity was 24 May, (42%, 6/14) which was approximately halfway through the three-month sampling period. Sampling on this day took place at the Rabuor slaughterhouse and all animals from this day originated from Kisumu County. Using this day as a case example, all positive animals (n = 6) had either walked (50%, 3/6) or arrived in a tuk-tuk (auto rickshaw), which is a small taxi built on a tricycle motorcycle body. Most (83%, 5/6) had waited only one day to be slaughtered and were indicated to be local animals from Kisumu. Interestingly, one of these local positive animals was less than two years old.

When we further examined the 78 seropositive animals as a sub-group, most seropositive animals (55%, 43/78) were said to have originated from Kisumu County, followed by Migori County (38%, 30/78). Nine of the seropositive animals were not purchased at a market and were assumed to be local resident animals. All animals from Migori County, except for one, were purchased at the Subakuria market (n = 29), and others at Ahero (13%, 10/78), Sondu (9%, 7/78), and Mamboleo (8%, 6/78). Overall, 91% (839/924) of animals were purchased at a market. Our surveillance method at slaughter captured animals from 30 markets in total, and most supplied less than five animals during the entire sampling period. The highest rates of seropositivity were found in animals from Subakuria (23%), Ahero (11%), Sondu (17%), Mamboleo (17%), and Ewaso Nyiro (7%) markets.

Most animals arrived at the slaughterhouse by lorry (38%, 354/923), in a tuk-tuk (21%, 197/923), or walked (34%, 318/923). Arriving to the slaughterhouse in a tuk-tuk (*p* < 0.005) or walking (*p* = 0.20) was associated with lower seropositivity compared with lorry travel, which could indicate that high-risk animals arrived from further away. Finally, we estimated the age of each individual and identified that animals greater than five years old were more likely to be seropositive (*p* = 0.02).

### 3.2. Vector Abundance and Diversity Summary according to the Trapping Method

#### 3.2.1. Biogents (BG) Trapping

A total of 8,892 mosquitoes were collected in the BG trap. Of those, 56.1% (n = 4,989) were males and 43.9% (n = 3,903) were females. Most (97.6%, n = 8,683) of the mosquitoes were collected at Rabuor and the rest (2.4%, n = 209) at Mamboleo. Almost all (99.2%, n = 8822) collected mosquitoes were *Culex*, the rest were *An. gambiae* 0.5% (n = 42), *Ae. aegypti* 0.2% (n = 19), *An. coustani* 0.1% (n = 6), and *An. funestus* 0.03% (n = 3). Most (78.0%, n = 3,046) of the female mosquitoes were unfed, 2.8% (n = 110) were blood-fed, 4.3% (n = 166) were half-gravid, and 14.9% (n = 581) were gravid.

#### 3.2.2. Prokopack Aspirator Results

A total of 4,976 mosquitoes were collected with the Prokopack aspirator; 56% (n = 2781) were males and 44% (n = 2195) females. Most (80%; 3991) of the mosquitoes were collected at Rabuor and 20% (n = 985) at Mamboleo. The proportions of mosquitoes collected were 85% (n = 1869) *Culex*, 9% (n = 186) *An. gambiae*, 6% (n = 129) *An. coustani,* 0.4% (n = 9), *Ae. aegypti*, and 0.1% (n = 2) *An. funestus*. Out of all the 2195 female mosquitoes, 65% (n = 1,433) were unfed, 12% (n = 272) blood-fed, 6% (n = 124) half-gravid, and 17% (n = 366) gravid. At both sites combined, more (66%; n = 1444) female mosquitoes were collected indoors than outdoors (34%, n = 751). At Mamboleo alone, more (67%, n = 346) female mosquitoes were collected outdoors than indoors (33%, n = 167). However, in Rabuor alone, more (76%, n = 1,277) mosquitoes were collected indoors than outdoors (24%, n = 405). The average numbers of *An. gambiae* and *An. coustani* female mosquitoes and cattle, goats, and sheep recorded at the time of sampling were significantly higher in Mamboleo than in Rabuor (Table 2). However, there were significantly more *Culex* female mosquitoes in Rabuor than Mamboleo (*p* < 0.001). At each slaughterhouse individually, there was minimal variation between sampling days each week.

#### 3.2.3. Ovitraps

A total of 64 ovitrap samplings for *Ae. aegypti* eggs were made in four alternate weeks per site. One specific ovitrap located in a metal structure not in use but adjacent to places where people buy and sell organs from the slaughterhouse collected the most eggs (80%, n = 520). Out of the 64 ovitrap samplings, 80% (n = 51) were in good condition and the remaining were either fallen (n = 7), lost (n = 3), or dry (n = 1). A total of 655 *Ae. aegypti* eggs were collected from the ovitraps: 53% (349/655) at Mamboleo and 7% (306/655) at Rabuor. The mean density of the *Ae. aegypti* eggs was not significantly different between Rabuor with 9.56 (CI 95%: 0.54–18.59) and Mamboleo with 10.91 (CI 95%: −0.77–22.59) (*p* = 0.85). Eggs were collected in all four weeks except for week three at Mamboleo. The majority (54.5%, n = 357) of eggs were collected in week two of sampling.

#### 3.2.4. Lateral Flow Test Results and Confirmatory PCR

In the initial experiments, it was determined that the maximum number of mosquitos that could be combined in a pool for the strip wicking to be successful was 10 mosquitos. We tested a total of 355 field-caught mosquito homogenate pools. Of these, 21 revealed a positive result (two lines) on the test strip. A random selection of 82 pools, including all suspect positives from the lateral flow assay, were all confirmed to be negative by RT-PCR. Thus, all the field samples positive according to the lateral flow assay were false positives, and we had a false positive rate of 6%.

### 3.3. Analysis of Mosquito Bloodmeals

A total of 44 blood-fed mosquitos were batched into 14 sample pools each containing from two to six blood-fed individuals (Table 3). These 14 pools were the only mosquito pools that were sent for bloodmeal analysis. Rarefaction curves were visualized for vertebrate species identified from the mosquito blood meal; the *x*-axis displays the number of reads (sequencing effort) and the *y*-axis represents species richness estimated through amplicon sequence variants (Figure 1).

We visualized boxplots of alpha diversity metrics for vertebrate species aggregated according to mosquito species and the study site, which revealed significant differences in vertebrate species between the mosquito species (Kruskal–Wallis observed *p* = 0.591, and Shannon *p* = 0.8409) and the study sites (observed *p* = 0.08225, and Shannon *p* = 0.5355), even though *Culex* spp. and the Mamboleo site appeared the richest.

A summary of the relative abundance of vertebrate hosts is presented in Table 4. Overall, a total of 14 vertebrate species were detected and belonged to two classes, i.e., Mammalia (82.3% relative abundance) and Aves (17.7%). *Culex* spp., by far, had the most diverse vertebrate blood meals (Table 4). However, when non-metric multidimensional scaling plots using Bray–Curtis dissimilarity were visualized, the study site, rather than the mosquito species, was the main driver of observed differences in the data (Figure 2). The study site explained 51.9% of variation in identified vertebrate hosts.

## 4. Discussion

We did not identify any acute cases over the high-risk sampling period during which animals entered Kisumu city from across Western Kenya; however, this region of Kenya has previously been described to be at low risk of RVFV [40]. Nonetheless, this study contributes to a more complete understanding of RVFV disease ecology at urban slaughterhouses through describing vectors involved in theoretical transmission and amplification in the urban environment. We also provide evidence that our previously described urban slaughterhouse sampling framework [8] can be successfully adapted to other slaughterhouses with different equipment and flow of animals. Future studies may investigate the potential of this same sampling framework in urban centers within high-risk regions. Not identifying any acute cases could additionally be explained by not sampling at an appropriate time, the level of regional risk in Western Kenya not extending into our catchment area, or other factors that influence the risk of an outbreak that the early warning model did not capture. Indeed, RVF outbreaks were later identified in Burundi in April 2022 and in Uganda in March 2023 [41,42] but not in Kenya. In this study, we identified an 8.5% overall livestock seroprevalence in this sampling effort and previously identified a 9% overall prevalence in October to November 2021, which could indicate that the risk was stable in this region and the original sample size (n = 304) was adequate to capture the prevalence and catchment area. There were also insignificant differences between overall prevalence at the two urban slaughterhouses, which suggests that the risk was influenced more by the animals’ purchase market and origin.

This slaughterhouse-based sampling method relies on accurate reporting from the stakeholder who has purchased the animal and assumes the truth of the information regarding the purchase market, transport means, and herd size. We identified six animals that had prior exposure to RVFV and seemed to be local animals residing in Kisumu as they walked to slaughter and were not purchased at a market, which further corroborated their reported origin. Unless these animals were purchased from the surrounding rural area after they were exposed, this may be initial proof of urban/peri-urban transmission in the Kisumu city area. In other studies, sentinel herds have provided evidence of ongoing hidden transmission [43], and this could be another method of monitoring peri-urban and urban involvement during outbreaks. To extend this effort into human health risk monitoring, previous studies carried out in Western Kenya highlighted a much higher rate of exposure in slaughterhouse workers compared with the general community [3]. Slaughterhouse workers in urban areas could therefore be the ideal population to monitor alongside livestock and vector surveillance, again displaying the profound potential of slaughterhouses to be a central point of One Health surveillance.

Our vector-trapping efforts at the same places we were testing livestock allowed us to assess how environmental factors can influence potential transmission dynamics. We collected data on the diversity and abundance of mosquito vectors from two different slaughterhouses with different ecological layouts and host presence. For example, Rabuor slaughterhouse was noted to have much more open drainage of water and blood mixture from the slaughterhouse floor to the blood disposal pit compared with Mamboleo, and animals were not congregated and held overnight before slaughter. At Rabuor slaughterhouse, we collected an extremely large number of *Culex* spp. which had the most diverse blood meals, and these differences are likely to be explained by the layout of the slaughterhouse. *Culex* spp. fed almost equally on ruminants and humans, indicating that if they were to become amplifying vectors for RVFV, human spillover could occur via infected mosquito bites This would be extremely difficult to tease out in recall surveys of slaughterhouse workers as they would also have daily direct exposure risks. We also collected a total of 655 *Aedes aegypti* eggs, which could be supportive of the urban slaughterhouse setting being a key location where infected eggs could lie dormant and later re-emerge. Other studies have also found very large numbers of *Aedes* spp. eggs and adult mosquitos in urban settings [29]. In our study, *Aedes* spp. fed primarily on cattle, and this is likely to have been influenced by the placement of our BG trap inside the cattle holding pen. Despite this interaction between *Aedes* spp. and cattle, as alluded to above and in our previous research, the greatest risk of outbreak initiation is likely to be from introduction of an inapparently infected animal. As the slaughterhouses we trapped at were larger with higher volumes of animals, our findings are likely to be unique to the urban setting, yet there were no species identified in the blood meals that cannot also exist in a rural setting. The rural, peri-urban, and urban interface is indeed complex, and with our study design, we cannot compare our results directly with rural sites. Furthermore, even though urban slaughterhouses have similarities, 52% of the variation in bloodmeal data was explained according to the study site, which highlights how landscape and drainage have profound influence on the diversity and abundance of mosquitos (Figure 2).

This study also tested the practical usage of a novel lateral flow assay, the VecTOR test, in a field setting. We did not detect any acute cases in the field. This assay was previously validated on known positive samples from the 2007 outbreak in Kenya, and the fact that we obtained a low number of false positives (6%) is promising for using this assay to monitor vector infections during interepidemic periods.

Limitations of this study include a sample size for livestock testing that was smaller than the disease freedom calculation, which limited our ability to reject the null hypothesis that there were no acute cases of RVFV in this urban slaughterhouse setting. The secondhand information we received from business stakeholders could be skewed and efforts to randomly corroborate their reports would strengthen data collection. Finally, the vector-trapping component of this study had limitations including different trap placement methods at each slaughterhouse. Future studies may also consider the benefit of combined mosquito trapping at urban and rural sites over the same period. We carried out bloodmeal analysis on only a portion of the blood-fed mosquitos, which could have inadvertently overrepresented some species in the distribution. The methods described above are excellent for understanding the total species involved in blood meal, but do not reliably confirm the proportion of the species as the approach relies on viable host DNA in the sample, and this effect could have been exacerbated by the unequal number of blood-fed mosquito species submitted from each slaughterhouse site. Nonetheless, given the similar results between the two analytical methods to interpret the bloodmeal results, we are confident that the vertebrate hosts identified in mosquito blood meals are representative of feeding patterns at urban slaughterhouses around Kisumu, Kenya.

Overall, as RVFV is an arbovirus with many host–vector relations and can infect many different mammalian hosts, the sampling effort described here could be adapted to other areas where domestic ruminants congregate, to determine urban viral amplification potential. This study and our previous work highlight that urban risk extends well beyond the household level, which differs from the results of other RVFV studies in rural areas of endemic countries where there is near 100% animal ownership. This expanded understanding of disease ecology in the urban setting may be more relevant to other non-endemic countries without expansive smallholder farmer systems or many household-level livestock owners. It would be helpful to use these data to quantify the potential for RVFV urban amplification at slaughterhouses and monitor risk dynamically with updated vector data and slaughterhouse stocking densities between seasons. The complex epidemiology of RVFV requires an integrated response, and urban slaughterhouses could ideally serve the purpose for continuing to expand understanding of urban RVFV transmission.

## 5. Conclusions

In conclusion, we did not identify any evidence of acute RVFV over this sampling period, although this slaughterhouse-based livestock surveillance system could be adapted to other slaughterhouses and urban areas to dynamically monitor risk of RVF and potentially other zoonoses. An accurate point-of-care diagnostic for RVFV that could also be used in livestock would have great potential in this setting. Slaughterhouses are diverse places for vector dynamics and those in the current study had several mosquito species that could contribute to theoretical transmission. By and large, *Culex* spp. were the most abundant and this is likely to be explained by the poor drainage at Rabuor slaughterhouse. *Aedes aegypti* also have a high level of breeding activity at slaughterhouses and the potential for vertical transmission should not be discounted. Trapping vectors at these ideal interfaces for disease exchange has allowed us to capture a new dimension of the diversity of risk in an urban setting. We encourage new efforts to describe risk in urban settings and site-specific testing and trapping to parameterize future models.

## Figures and Tables

**Figure 1 pathogens-13-00488-f001:**
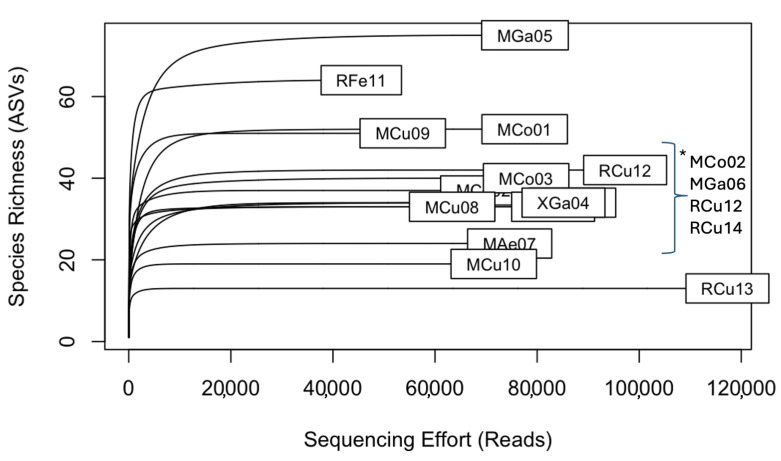
Rarefaction curves for vertebrate species identified from mosquito blood meals collected at Rabuor and Mamboleo, Kisumu. The *x*-axis is sequencing effort, and the *y*-axis is species richness. The tips of each curve are labeled with the mosquito pool identity. * Unreadable pool names overlapped.

**Figure 2 pathogens-13-00488-f002:**
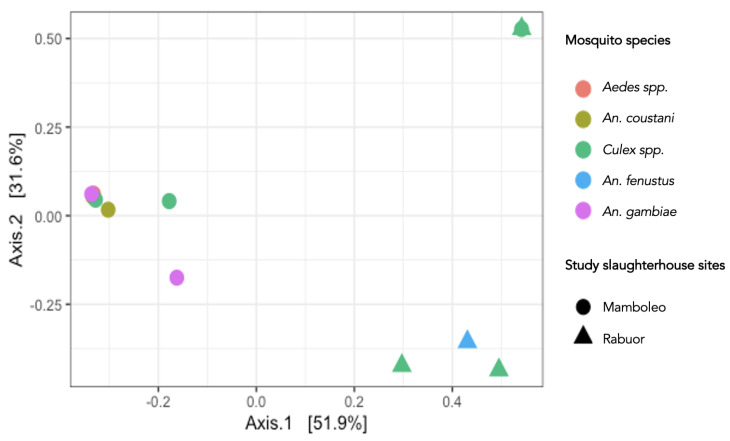
Non-metric multidimensional scaling (nMDS) plots of vertebrate host metabarcodes associated with mosquito species collected from Rabuor and Mamboleo sites in Kisumu, Kenya. Mosquito species are indicated by color i.e., *Aedes* spp. (orange), *An coustani* (olive), *Culex* spp. (green), *An fenustus* (blue), and *An gambiae* (purple), whereas the study sites are indicated by shapes i.e., Mamboleo (solid circle) and Rabuor (solid triangle). The plots are based on Bray–Curtis dissimilarity distances and show how similar the sampling units are. The closer the units are to each other on the plot, the more similar they are. In this case, the study site rather than the mosquito species is the main driver of observed differences and explains 51.9% of the variation of the vertebrate hosts identified.

**Table 1 pathogens-13-00488-t001:** Descriptive statistics of RVF IgG antibody results of livestock sampled at slaughter.

Predictor	Variable	Total (n = 923)	Total Seropositive (n = 78)	Seropositive Rate (%)	*p*-Value
Slaughterhouse	Mamboleo	503	42	8.3	0.90
	Rabuor	420	36	8.6	
Species	Sheep *	313	8	2.6	<0.001
	Goat	285	6	2.1	
	Cattle	325	64	19.7	
Days in holding for slaughter	1	412	33	8.0	0.51
	2	171	11	6.4	0.20
	3	146	7	4.8	0.06
	4	81	12	14.8	0.04
	5	46	8	17.4	0.51
	6	36	4	11.1	0.74
	7+ *	31	3	9.7	
Herd size, continuous [range: 1–100]	Median = 10				0.02
	1st Qu = 4				
	3rd Qu = 20				
Market purchase?	Yes	839	69	8.2	0.43
	No	84	9	10.7	
Major purchasing markets	Ahero *	93	10	10.8	
	Chiga	77	3	3.9	0.11
	Kipsitet	58	1	1.7	0.07
	Rabuor	155	6	3.9	0.04
	Subakuria	128	29	22.7	0.03
Transport means	Lorry *	354	43	12.1	
	Motorcycle	50	0	0	0.98
	Pickup truck	4	0	0	0.99
	Tuk-tuk	197	6	3.0	<0.005
	Walking	318	29	9.1	0.20
Estimated age (years)	Less than 2 *	115	10	8.7	
	2	81	2	2.5	0.11
	3	223	12	5.4	0.27
	4	224	13	5.8	0.35
	5 +	203	41	20.2	0.02

* Reference for categorical predictors. Qu: quartile.

**Table 2 pathogens-13-00488-t002:** Comparison between Rabour and Mamboleo slaughterhouses of the female mosquitoes collected with the Prokopack aspirator, people, cattle, goats, and sheep.

Species	Slaughterhouse	Average (95% CI)	ANOVA	*p*-Value
*Aedes aegypti*	Rabuor	0.1 (0.0–0.2)	F_1,78_ = 0.1	0.76
	Mamboleo	0.1 (0.0–0.3)		
*Anopheles gambiae*	Rabuor	0.7 (0.3–1.0)	F_1,78_ = 17.1	<0.001
	Mamboleo	4.0 (2.4–5.6)		
*Anopheles funestus*	Rabuor	0.0 (0.0–0.0)	F_1,78_ = 1.0	0.32
	Mamboleo	0.05 (−0.05–0.15)		
*Culex* spp.	Rabuor	41.3 (28.0–54.6)	F_1,78_ = 29.4	<0.001
	Mamboleo	5.4 (3.7–7.1)		
*Anopheles coustani*	Rabuor	0.0 (0.0–0.0)	F_1,78_ = 26.2	<0.001
	Mamboleo	3.2 (2.0–4.5)		
People seen at sampling	Rabuor	36.0 (29.6–42.4)	F_1,38_ = 1.3	0.23
	Mamboleo	40.1 (36.1–44.1)		
Cattle seen at sampling	Rabuor	2.1 (1.4–2.8)	F_1,38_ = 335.1	<0.001
	Mamboleo	61.9 (55.1–68.6)		
Goats seen at sampling	Rabuor	5.1 (3.1–7.1)	F_1,38_ = 58.3	<0.001
	Mamboleo	30.4 (23.8–37.1)		
Sheep seen at sampling	Rabuor	17.4 (13.4–21.3)	F_1,38_ = 36.3	<0.001
	Mamboleo	49.4 (39.0–59.8)		

**Table 3 pathogens-13-00488-t003:** Summary of mosquito pools by site and species for bloodmeal analysis.

Slaughterhouse Collected	Mosquito Species	n=	Name Assigned to Pool *
Mamboleo	*Anopheles coustani*	4	MCo01
Mamboleo	*Anopheles coustani*	4	MCo02
Mamboleo	*Anopheles coustani*	4	MCo03
Both	*Anopheles gambiae*	6	XGa04
Mamboleo	*Anopheles gambiae*	5	MGa05
Mamboleo	*Anopheles gambiae*	5	MGa06
Mamboleo	*Aedes aepypti*	2	MAe07
Mamboleo	*Culex* spp.	2	MCu08
Mamboleo	*Culex* spp.	2	MCu09
Mamboleo	*Culex* spp.	2	MCu10
Rabuor	*Anopheles fenestus*	2	RFe11
Rabuor	*Culex* spp.	2	RCu12
Rabuor	*Culex* spp.	2	RCu13
Rabuor	*Culex* spp.	2	RCu14

* Corresponds to name assignment in figures below.

**Table 4 pathogens-13-00488-t004:** Relative frequency of vertebrate hosts identified in different mosquito species.

			*Aedes* spp.		*Anopheles coustani*		*Culex* spp.		*Anopheles gambiae*		*Anopheles fenestus*	
Study Site	Species Scientific Name	Species Common Name	n=	Freq	n=	Freq	n=	Freq	n=	Freq	n=	Freq
Mamboleo	*Bos taurus*	Cattle (domestic)	73,832	0.989	219,708	0.979	111,526	0.601	139,379	0.851		
	*Turdus pelios*	African thrush bird			3		61,546					
	*Homo sapiens*	Human	102	0.001	4285		1453		19,373	0.118		
	*Gallus gallus*	Red junglefowl					9878					
	*Caprap hircus*	Goat (domestic)			14	0.000	0		4504	0.027		
	*Bos javanicus*	Cattle (banteng)	677	0.009	470	0.002	786		543	0.003		
	*Bos indicus x Bos taurus*	Cattle (zebu) x Cattle (domestic)	7	0.000			235					
	*Gallus sonneratii*	Grey junglefowl					57					
	*Homo heidelbergensis*	Human							16	0.000		
	*Capra caucasica*	West Caucasian tur goat							11	0.000		
	*Ovis orientalis*	Sheep			4	0.000						
	*Capra aegagrus*	Wild goat							7	0.000		
	*Bos frontalis x Bos taurus*	Cattle (gayal) x Cattle (domestic)							3	0.000		
Rabuor	*Bos taurus*	Cattle (domestic)					14,769	0.050			853	0.019
	*Turdus pelios*	African thrush bird					118,670	0.399			63	0.001
	*Homo sapiens*	Human					164,127	0.551			44,635	0.980
	*Gallus gallus*	Red junglefowl					29	0.000				
	*Bos javanicus*	Cattle (banteng)					56	0.000				

## Data Availability

The datasets used and analyzed during the current study are available from the corresponding author on reasonable request.

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
