# Peer review of "Expanding Understanding of Urban Rift Valley Fever Risk and Associated Vector Ecology at Slaughterhouses in Kisumu, Kenya"

_pathogens, 2024, doi:10.3390/pathogens13060488_

Round 1

Reviewer 1 Report

Comments and Suggestions for Authors

General comments:

The authors performed field sampling of animal blood, adult mosquito as well as mosquito progeny at two urban slaughter houses, and assessed the risk of RVFV introduction by testing for viral prevalence serologically or genetically. These authors have previously reported an urban slaughter house sampling framework, and proposed that the influx of livestock might facilitate the RVFV introduction to the urban area. The current study was initiated in respond to a RVF alert by AFO in early 2022 when the risk of RVF outbreaks was high due to ideal weather conditions.

In the current study, these authors picked two slaughter houses including the largest slaughter facility in the region. However, no clue of acute viral infection was observed by checking the anti-RVF IgG and IgM antibodies in the animal serum samples. The presence of viral genome in the mosquito samples is also proved to be negative by pooled PCR. The lack of viral infection was explained by not sampling at an appropriate time, or a relatively short sampling period (two months). This lowers the value of the report since no direct evidence support the idea of urban slaughter houses indeed facilitate the transmission of RVF. In addition to viral detection analysis, the authors also provided information of vector species abundance, blood types taken by the mosquitoes as well as evidence of Aedes mosquito reproduction at the two slaughter houses.

Although no direct evidence showing that the slaughter houses contribute to the outbreak of RVF, the current study provided a feasible sampling frame work in slaughter facilities for zoonoses surveillance purpose. Plus, the serum and vector data generated in the study is a good addition to the current understanding of RFV circulation in the endemic regions. 

Minor comments:

1.The legend of figure 2 is not well displayed.

2. The x and y axis title need to be more specific for Figure 2.

3. Line 24, “...than any other virus of medical importance” Better to rephrase the sentence since the interaction analysis by Yee and colleagues (2022) did not include all the virus class of medical importance.

4. Line 379, check the percent and the number.

Author Response

Dear reviewer one, 

Thank you kindly for your time reviewing our manuscript. I've addressed the minor points you brought up and summarise the changes below. 

1.The legend of figure 2 is not well displayed.

Thank you for catching this mistake. The figure has been corrected on line 444

  1. The x and y axis title need to be more specific for Figure 2.

Thanks for this comment. These mon-metric multidimensional scaling (nMDS) plots are somewhat unique in that the axes do not represent one variable. This is the typical way of displaying plots that compare multiple factors on one graph which is common for blood meal analyses in mosquitos. I've added some text to the end of the figure legend to help make this point more clear. 

of vertebrate host metabarcodes associated with mosquito species collected from Rabuor and Mamboleo sites in Kisumu, Kenya. Mosquito species are indicated by color i.e., Aedes spp (orange), An coustani (olive), Culex spp (green), An fenustus (blue), and An gambiae (purple), whereas the study sites are indicated by shapes i.e., Mamboleo (solid circle), and Rabuor (solid triangle). The plots are based on Bray-Curtis dissimilarity distances and show how similar the sampling units are. The closer the units are to each other on the plot, the more similar they are. In this case, the study site rather than the mosquito species is the main driver of observed differences and explained 51.9% of the variation of the vertebrate hosts identified. Essentially, the closer the points are on the graph the more they are related and the axis does not have a name. 

  1. Line 24, “...than any other virus of medical importance” Better to rephrase the sentence since the interaction analysis by Yee and colleagues (2022) did not include all the virus class of medical importance.

Great point, I have rephrased this to tone down the senstence. 

  1. Line 379, check the percent and the number. 

Thank you, there was a mistake with a missing 7, should be 76%. Confirmed it is now correct. 

Please let me know if you identify any further corrections needed, 

Keli 

Reviewer 2 Report

Comments and Suggestions for Authors

This is an intersting paper as its examines the transmission dynamics of Rift Valley Fever virus in an urban setting and uses slaughterhouse data to distinguish between rural and urban areas.  The other strength is to compare slaughterhouse data with the information on the mosquito vector.

The paper can be improved by the authors more carefully explaining their the nature of the pools tested.  Only a sample of postive pools were examined using PCR and  it was not clear how many mosquitoes were in each pool although it was stated that 10 were sufficient for a pool.  This needs to be clarified.

The study used sequencing to identify the species from the blood meal of the female mosquitoes. This data can be further supported in the discussion with some background on the population density of the species in the urban setting and if the species diversity in the blood meal is indicative of urban transmission vs rural.

Comments on the Quality of English Language

The English appears to be written by native English Speakers and requires very little editing.

Author Response

Dear reviewer two, 

Thank you kindly for the comments.

Regarding the pools tested for blood meals:

We only tested a subset of the mosquitos that were fully blood fed. The pools described with up to 10 mosquitos was for the lateral flow assay to detect RVFV. A summary of the pools sent for blood meal analysis is summarised in table 4, but I have added additional text in the methods to make this point more clear for our future readers.  

Re discussion on the population density of the species in the urban setting and if the species diversity in the blood meal is indicative of urban transmission vs rural: 

This is another great point, but unfortunatly with this study desgin we can't compare directly to the rural setting and there are no other studies trapping mosquitos at rural slaughterhouses. I've added this as a limitation of the study and pointed towards the overlap between the urban, peri-urban, and rural interface. Future studies could place traps at the same time in the urban and rural settings. Hopefully this point has been addressed adequately. 

Thank you for your time and please let me know if additional modifications are required. 

Keli